# 7-Hydroxy Frullanolide Ameliorates Isoproterenol-Induced Myocardial Injury through Modification of iNOS and Nrf2 Genes

**DOI:** 10.3390/biomedicines11092470

**Published:** 2023-09-06

**Authors:** Saif Ullah, Taseer Ahmad, Muhammad Ikram, Hafiz Majid Rasheed, Muhammad Ijaz Khan, Taous Khan, Tariq G. Alsahli, Sami I. Alzarea, Musaad Althobaiti, Abdul Jabbar Shah

**Affiliations:** 1Cardiovascular Research Group, Department of Pharmacy, COMSATS University Islamabad, Abbottabad Campus, University Road, Abbottabad 22060, Pakistan; saifullahsarim@gmail.com (S.U.); ikram@cuiatd.edu.pk (M.I.); taouskhan@cuiatd.edu.pk (T.K.); 2Department of Pharmacology, College of Pharmacy, University of Sargodha, University Road, Sargodha 40100, Pakistan; taseer.ahmad@uos.edu.pk; 3Division of Clinical Pharmacology, Department of Medicine, Vanderbilt University Medical Center, Nashville, TN 37232, USA; 4Faculty of Pharmacy, The University of Lahore, Lahore 54590, Pakistan; drmajid.rph@gmail.com; 5Greater Baltimore Medical Center, Towson, MD 21204, USA; mikhan1502@gmail.com; 6Department of Pharmacology, College of Pharmacy, Jouf University, Sakaka 72341, Saudi Arabia; tgalsahli@ju.edu.sa (T.G.A.); samisz@ju.edu.sa (S.I.A.); 7Department of Pharmacology and Toxicology, College of Pharmacy, Taif University, Taif 21944, Saudi Arabia; md.althobaiti@tu.edu.sa

**Keywords:** 7-hydroxy frullanolide, myocardial infarction, isoproterenol, oxidative stress, Nrf2, iNOS

## Abstract

Myocardial infarction (MI) is the principal cause of premature death. Protecting myocardium from ischemia is the main focus of intense research. 7-hydroxy frullanolide (7-HF) is a potent anti-inflammatory agent, showing its efficacy in different acute and chronic inflammatory disorders such as atherosclerosis, suggesting it can be a potential cardioprotective agent. For the induction of MI, Sprague–Dawley rats (n = 5) were administered isoproterenol (ISO) 85 mg/kg s.c at 24 h intervals for two days. The potential cardioprotective effect of 7-HF and its mechanisms were explored by in vivo and in vitro methods. 7-HF significantly prevented the extent of myocardial injury by decreasing the infarct size, preserving the histology of myocardial tissue, and reducing the release of cardiac biomarkers. Further, 7-HF increased the mRNA expression of cardioprotective gene Nrf2 and reduced the mRNA expression of iNOS. 7-HF also improved cardiac function by decreasing the cardiac workload through its negative chronotropic and negative ionotropic effect, as well as by reducing peripheral vascular resistance due to the inhibition of voltage-dependent calcium channels and the release of calcium from intracellular calcium stores. In conclusion, 7-HF showed cardioprotective effects in the MI model, which might be due to modulating the expression of iNOS and Nrf2 genes as well as improving cardiac functions.

## 1. Introduction

Myocardial infarction (MI) is the most frequent form of Ischemic Heart Disease (IHD), accounting for over 15% of mortality each year [1]. IHD, is the term given to heart diseases caused by narrowed coronary arteries resulting in a reduced supply of blood to the heart muscle [2]. Globally, about 32.4 million cases of MI and stroke are reported each year [3]. Clinically, during MI, an acute myocardial injury is detected by the rise in cardiac biomarkers secondary to acute myocardial ischemia [4]. Myocardial infarction is followed by various pathophysiological and biochemical alterations like ulceration or rupture of atheroma, thrombosis, occlusion of coronary artery resulting in decreased supply of oxygen, which stimulates lipid peroxidation (LPO), generation of reactive oxygen species (ROS) and decline in the nitric oxide (NO) level leading to pathological changes in the myocardium [5]. It has also been studied that oxidative stress produced by free radicals is associated with the decline in the antioxidants defense system like catalase (CAT), superoxide dismutase (SOD), glutathione peroxidase (GPx) due to the down-regulation of nuclear factor (erythroid-derived 2)-related factor 2, also known as NFE2L2 or Nrf2. Nrf2 is a major transcription factor that is essential in inducing endogenous antioxidant enzymes in response to oxidative stress. Additionally, Nrf2 also mediates cellular repair and regeneration due to its anti-tumor, anti-inflammatory and antiapoptotic effects [6,7]. The pathophysiological variations also include activation of iNOS activity in cardiomyocytes which play a crucial role in the induction of cardiomyocytes apoptosis [8], overexpression of calcium channels increasing intracellular Ca^2+^ levels [9] and elevated cardiac biomarkers cardiac troponin I (cTnI), creatine kinase myocardial band (CK-MB), lactate dehydrogenase (LDH), alanine aminotransferase (ALT) and aspartate aminotransferase (AST) [10,11].

Protecting myocardium from ischemia is the main focus of intense research. However, despite various publications and different successful preclinical studies, so far, no effective cardioprotective drug has reached the phase of clinical trials [12]. Therefore, a perpetual search for alternative cardioprotective agents is in progress. The extract and phytochemical constituents of many medicinal plants have been evaluated as cardioprotective, acting via different mechanisms [13]. Different sesquiterpene lactones derived from plant sources such as parthenolide have been studied for their cardioprotective effects [14]. 7-hydroxy frullanolide (7-HF), a sesquiterpene lactone, derived from *Sphaeranthus indicus*, has been studied to suppress the production of inflammatory cytokines (e.g., TNF-α, IL-6) resulting in marked protection in acute and chronic inflammatory diseases, such as atherosclerosis, arthritis and colitis [15,16,17]. Furthermore, 7-HF was discovered to modulate the expression of numerous bioactive proteins associated with various components of metabolic syndrome. 7-HF has also been studied to inhibit the accumulation of lipids in high glucose-induced macrophage cells of an in vitro model of atherosclerosis by suppressing the protein expression of CD36 [18]. Therefore, CD36 is suggested to be an essential risk factor for cardiovascular disease and a potential maker of atherosclerosis. Studies have also found that inhibition of CD36 by its ligands and its ablation from cardiomyocytes protects cardiac tissue from post-ischemic damage and improves functional recovery of the myocardium [19,20].

However, no study has been reported up to date on the cardioprotective effect of 7-HF. Hence, the current study is intended to explore the potential protective effect of 7-HF on isoproterenol-induced MI in rats. We hypothesized that 7-HF would be a novel strategy for protecting against ISO-induced cardiac injury by modifying the expression of Nrf2 and iNOS genes and by reducing the cardiac workload. To test this hypothesis, we evaluated the effect of 7-HF on the mRNA expression of these genes, antioxidant enzymes, and isolated rat atria and aorta. The results of this study may provide a new mechanistic insight into the protective actions of 7-HF against ISO cardiotoxicity, as well as a potential strategy for the prevention of ischemic heart disease.

## 2. Methods

### 2.1. Chemicals and Reagents

Standard drugs like isoproterenol hydrochloride, potassium chloride, atenolol, atropine sulfate, Nω-nitro-L-arginine methyl ester (L-NAME), phenylephrine hydrochloride, indomethacin sodium, angiotensin II (Ang II) and verapamil hydrochloride were obtained from Sigma-Aldrich (St. Louis, MO, USA) and dimethyl sulfoxide (DMSO) from Alfa Aesar, Heysham, Lancaster, UK. Further standard drugs, including 2-aminopyridine (2-AP), tetraethylammonium (TEA), barium chloride, and thiopental sodium injections, were purchased from Santa Cruz Biotechnology, Dallas TX, USA. All drugs were solubilized by normal saline except indomethacin, which was dissolved in 5% ethanol. The test compound 7-hydroxy frullanolide was first dissolved in 5% DMSO for in vivo and 3% DMSO for in vitro studies. TEA was dissolved into Kreb–Henseleit solution.

### 2.2. Source of 7-Hydroxy Frullanolide (7-HF)

The compound 7-HF was isolated from the chloroform fraction of *Sphaeranthus indicus* by column chromatography using n-hexane and ethyl acetate as solvent systems. It was purified by repeated column chromatography and characterized with mass spectrometry and nuclear magnetic resonance (NMR) [21].

### 2.3. Ethical Approval

Experiments were conducted according to the guidelines of the Institute of Laboratory Animal Resources, Commission on Life Sciences, National Research Council (NRC, 2011) after the approval by the ethical committee of the Pharmacy Department, COMSATS University Islamabad, Abbottabad Campus, Pakistan (protocol number, 4470/21/3136 approved on November 2021).

### 2.4. MI Induction in Rats

Isoproterenol (ISO) is a synthetic catecholamine used to induce MI in rats. ISO 85 mg/kg dissolved in saline solution was administered s.c for 2 days at an interval of 24 h.

### 2.5. Animals and Experimental Design

Sprague–Dawley (SD) rats of either gender weighing 200–240 g were kept at the animal house of the COMSATS University Islamabad (CUI), Abbottabad campus under hygienic conditions and a controlled environment maintained at a temperature of 22–25 °C and the light-dark cycle of 12 h. Rats were provided water and food *ad libitum*.

Rats were randomly categorized into 6 groups, containing five rats in each group. Group I, the normal control group, was administered 3% DMSO for 10 d and normal saline (10 mL/kg/d) on d 9 and 10. Group II, the negative control group, received ISO (85 mg/kg/d i.p) on d 9 and 10. Group III, positive control group, was given atenolol (10 mg/kg/d p.o) for 10 d and isoproterenol (85 mg/kg/d i.p) on d 9 and 10. Groups IV, V, and VI were given 7-hydroxy frullanolide in three different doses (5, 10, 25 mg/kg/d p.o) for 10 days and ISO (85 mg/kg/d) on d 9 and 10. The rats were anesthetized using thiopental sodium (65 mg/kg, i.p) 12 h after the last dose of ISO.

### 2.6. Electocardioghrapy

For ECG measurement, rats were administered thiopental sodium (65 mg/kg, i.p) after complete anesthesia, and the acupuncture needle electrodes were inserted s.c in the pattern of lead II (right foreleg = −ve electrode, left foreleg = ground electrode, and left rear leg = +ve electrode) of the ECG plot (Figure 1). ECG was performed using Powerlab attached with BioAmp and interpreted by LabChart 7 software (ADInstruments, Australia). The amplification of each channel was kept at the rate of 2 kHz and 5 mV range of a high-pass filter setting of 1 Hz. Changes in the pattern of ECG were recorded and analyzed [22].

### 2.7. Myocardial Infarct Quantification

The infarct size was quantified by the tissue enzyme staining technique. The infarcted tissues were stained using 1% solution of 2, 3, 5-triphenyl tetrazolium chloride. Heart of rats in all groups were exsected and frozen immediately at −20 ℃ for 1 h. A 3–5 mm thick slice from the heart tissues was then obtained by making a cut parallel to the AV groove from the apex to the base. The tissue slices were then incubated for 20 min in 2, 3, 5-triphenyl tetrazolium chloride 1% solution at pH 7.4 and 37 °C temperature. After 20 min, the tissues were fixed in 10% saline for 30 min more. The tissues were then photographed, and ImageJ software (Version; v1.53k, Wayne Rasband and contributors, National Institute of Health, USA) was used to calculate the infarcted area [23].

### 2.8. Histopathological Evaluation

For the histopathological study, heart tissues of experimental rats were isolated and fixed immediately in a 10% solution of buffered formalin. For slide formation, the tissue slice was made by cutting the heart from the apex to the base parallel to the AV groove. The slice was dehydrated by different concentrations of ethyl alcohol and cleaned with xylene subsequently before being placed in tissue molds containing paraffin. A thin slice of 5 µm was obtained and stained with hematoxylin and eosin. The histology of the tissue was examined under a light microscope for comparison and interpretation [24].

### 2.9. Estimation of Cardiac Markers

For estimation of cardiac markers, the blood sample was taken from rats by cardiac puncture 24 h after the administration of the last dose of ISO and centrifuged to separate the serum for analysis. The cardiac biomarkers such as cardiac troponin I and creatine kinase-MB (CK-MB) were analyzed from plasma by immunochromatography using Fluoro-Checker TM TRF, Nano-Ditech Corporation, USA). The levels of lactate dehydrogenase (LDH), alanine transaminase (ALT), and aspartate transaminase (AST) were measured by Spectrophotometry (Beckman Coulter—DxC 700 AU, Brea, CA, USA).

### 2.10. Determination of Oxidative Stress and Antioxidant Activity

The heart excised was rinsed with saline and froze immediately. The heart tissue was homogenized and centrifuged at 10,000× *g* at 4 °C. The clear supernatant obtained was used to determine the level of lipid peroxidation (MDA content), marker of oxidative stress, and antioxidative enzymes. The formation of malondialdehyde (MDA) or lipid peroxidation was estimated by the method of Slater and Sawyer [25]. Superoxide dismutase (SOD) was evaluated by the method of Mishra and Fridovich [26]. Catalase (CAT) was determined by the method of Aebi [27]. GSH was estimated by the method of Moron and Depierre [28].

### 2.11. PCR Analysis for iNOS and Nrf2

Total RNA was extracted from heart tissues using guanidine/phenol solution (biozol RNA isolation reagent) according to the manufacturer’s protocols. The concentration and purity of RNA was measured with a nanodrop (Titertek Berthold, Baden-Württemberg, Germany) instrument. Then, first-strand cDNA was obtained using 2 µg of total RNA according to the manufacturer protocol (WizScript cDNA synthesis kit, Seongnam, South Korea). The relative expression of mRNA was analyzed by preparing a PCR reaction mixture with DreamTaq Green PCR Master mix (2X) (ThermoScientific, Waltham, MA, USA), diluted cDNA along with gene-specific primers, and nuclease-free water to make the final volume up to 10 µL. The sequence of primers for specific genes is listed in Table 1. The housekeeping gene GAPGH was used to normalize the target gene expression. The expression level of genes was quantified using ImageJ software.

### 2.12. Principal Mechanism on Isolated Rat Atria

Rats were euthanatized by exsanguination from the heart under excess ether anesthesia using inhalant method, and atria were isolated and inclined in a tissue bath having Kreb’s solution aerated with carbogen gas (5% CO_2_ in O_2_), maintaining the temperature at 32 °C. The composition of Kreb’s solution was (mM): glucose 11.70, KCL 4.70, NaCl 118.20, MgSO_4_ 1.20, CaCL_2_ 2.50, KH_2_PO_4_ 1.30, and NaHCO_3_ 25.00. The tension of 1 g was given to each tissue and was allowed for 15–20 min to become stabilized. Afterward, stabilization alterations in the isometric tension of the atria were determined through a force–displacement transducer bridged with PowerLab Data Acquisition System. To evaluate the principal mechanism, right atria were pre-incubated for 30 min with atropine (1 µM) and β-blocker (1 µM), and the activity of 7-HF was determined in spontaneous tissue and in the presence of these blockers [29].

### 2.13. Vascular Activity on Isolated Rat Aorta

The descending thoracic aorta of the rat was excised and placed in Kreb’s solution, fats attached to the aorta were removed, and rings of 3 mm were formed. The aortic rings were placed cautiously in a tissue bath bearing Kreb’s solution, gassed with carbogen maintained at 37 °C, and attached with a force–displacement transducer coupled with PowerLab Data Acquisition System (ADInstruments, Sydney, Australia). A tension of 2 gm was given to the rings and stabilized for 60–90 min, with periodic changing of solution every 15 min. Some of the aortae were denuded by removing the endothelium. For the integrity of endothelium, the relaxing effect of acetylcholine was evaluated against phenylephrine-induced contraction [30].

#### 2.13.1. Effect of 7-Hydroxy Frullanolide on Phenylephrine, K^+^ (80 mM) and Ang II Pre-Contraction

The effect of 7-HF was determined on stable contraction produced by the vasoconstrictors like phenylephrine (1 µM), K^+^ (80 mM), and Ang II (5 µM) on aortic ring preparation. The experiment was performed following the Chan et al. 2006 protocol with some modifications [31]. Vasorelaxation was established as the percent of contractions produced by the agonist [32].

#### 2.13.2. Evaluating the Effect of 7-Hydroxy Frullanolide in L-NAME, Atropine and Indomethacin Presence

To evaluate the role of NO, muscarinic receptor, and prostacyclin, endothelium-intact aortic rings were pretreated with L-NAME (10 µM), atropine (1 µM), and indomethacin (1 µM) for 25 min before contraction with phenylephrine (1 µM). The cumulative concentration-response (CRC) of 7-HF was compared both in the presence and absence of these inhibitors [33,34].

#### 2.13.3. Effect of 7-Hydroxy Frullanolide on Voltage-Dependent Calcium Channel

To study the potential involvement of voltage-dependent Ca^2+^ channel, a series of experiments was performed for 7-HF and standard drug verapamil. The tissue integrity was confirmed and stabilized by inducing contraction with K^+^ (80 mM). To make tissue free of Ca^2+^, the solution of the organ bath was replaced with EGTA containing Ca^2+^-free Kreb’s solution, and washed the tissue 3–4 times. The solution was then replaced with Ca^2+^ free K^+^ enriched Kreb’s solution, and the tissue was stabilized for 20 min. Different concentrations of CaCl_2_ (0.01–10.0 mM) were added cumulatively in order to construct control CRCs. The tissues were then pre-incubated with different concentrations of 7-HF (3, 10, and 30 µM) for 40 min, and the CRCs were again constructed to evaluate the potential calcium channel blockage activity. The same experiment was repeated for the standard drug verapamil [31,35].

#### 2.13.4. Effect of 7-Hydroxy Frullanolide on Intracellular Ca^2+^ Stores

This experiment was performed to determine the possible inhibitory response of 7-HF on the release of Ca^2+^ from intracellular calcium stores. The aortic rings were initially treated with Ca^2+^-free Kreb’s solution for 20 min before the addition of phenylephrine. After the construction of the control curve induced by phenylephrine, the tissues were washed 3–4 times and incubated for 40 min with normal Kreb’s solution to refill the Ca^2+^ stores. The tissues were then incubated with Ca^2+^-free Kreb’s solution for 20 min. The next contraction was produced by phenylephrine (1 µM) after the tissue was pre-incubated with different concentrations of 7-HF for 40 min. A similar experiment was performed for the standard drug verapamil [36,37].

#### 2.13.5. Effect of 7-Hydroxy Frullanolide on K^+^-Channels

The effect of 7-HF was determined in the presence and absence of potassium channel blockers; tetraethylammonium (5 mM) and barium chloride (30 µM) were added 20 min before the phenylephrine-induced contraction. Different concentrations of 7-HF were added cumulatively to obtain CRC after the contraction reached its stable phase.

#### 2.13.6. Statistical Analysis

For the statistical analysis of the data, one and two-way ANOVA followed by Tukey’s and Bonferroni’s tests were used using GraphPad Prism (GraphPad Prism version 5.01, GraphPad software, San Diego, California USA). The values were expressed as mean ± SEM. *p-*values less than 0.005 were considered statistically significant.

## 3. Results

### 3.1. In Vivo Studies

#### Effect of 7-Hydroxy Frullanolide on Electrocardiograph (ECG) Pattern

Rats of the control group treated with 3% DMSO alone did not produce any significant alteration in the ECG pattern (Figure 2A). The ISO-alone treated rats showed significant ST-segment elevation, pathological Q wave, unequal PR interval, and tachycardia (Figure 2B). Rats pretreated with 5 mg/kg 7-HF showed a complete disappearance of pathological Q wave and a decrease in ST-segment elevation (Figure 2C). Rats pretreated with 10 mg/kg significantly reduced ST-segment elevation (Figure 2D). The ISO-induced alterations were almost completely renovated to normal in rats pretreated with 7-HF 25 mg/kg (Figure 2E) and atenolol 10 mg/kg (Figure 2F).

### 3.2. In Vitro Studies

#### 3.2.1. Effect of 7-Hydroxy Frullanolide on Infarct Size

Images of the ventricular slices stained with triphenyl tetrazolium chloride (TTC) revealed pronounced whitish TTC-negative regions in the ISO-alone treated hearts, showing the presence of infarction. The infracted areas did not stain with TTC and are diffusely located on the myocardium. In contrast to the ISO-alone treated group, the infarcted area was significantly reduced by the pretreatment of groups with 7-HF (5, 10 and 25 mg/kg) and atenolol (10 mg/kg) (Figure 3).

#### 3.2.2. Effect of 7-Hydroxy Frullanolide on Cardiac Biomarkers

As shown in Figure 4, a significant rise in the serum level of cardiac biomarkers of the group treated with ISO alone was observed. In comparison to the normal control group, administration of ISO alone significantly increased the level of cardiac biomarkers, cTnI (1.5803 ± 0.079), CK-MB (230 ± 6.81), LDH (616 ± 15.03), ALT (90.02 ± 2.30) and AST (120 ± 10.38). Rats that are given the standard drug atenolol (10 mg/kg) + ISO significantly reduced the release of the cardiac biomarkers induced by ISO, such as cTnI (0.684 ± 0.02), CK-MB (135.87 ± 6.84), LDH (376.57 ± 12.30), ALT (57.66 ± 2.76) and AST (60.09 ± 4.42).

Moreover, pretreatment with 7-HF (5 mg/kg) + ISO significantly reduced serum level of cTnI (1.23 ± 0.04) (Figure 4A) and CK-MB (193 ± 5.81) (Figure 4B); however, no significant alteration was found in the serum level of LDH (570.93 ± 28.32), ALT (81.22 ± 3.49) and AST (105.72 ± 4.14) (Figure 4C–E). Group pretreated with 7-HF (10 mg/kg) + ISO restored all cardiac biomarkers significantly, including cTnI (0.926 ± 0.03), CK-MB (176.27 ± 5.16), LDH (463.77 ± 17.903), ALT (58.80 ± 3.68) and AST (72.986 ± 7.37) (Figure 4A–E). These cardiac biomarkers were also significantly restored in group pretreatment with 7-HF (25 mg/kg) + ISO, cTnI (0.611 ± 0.01), CK-MB (134.53 ± 6.73), LDH (342.73 ± 9.15), ALT (39.75 ± 1.60) and AST (54.45 ± 4.01), compared to ISO-alone treated group (Figure 4A–E).

#### 3.2.3. Histopathological Examination of Heart Tissue

The histopathological evaluation of control rats showed normal myocardial cells with vesicular central oval nuclei, acidophilic cytoplasm, and cardiac fibers separated by interstitial connective tissue with flat nuclei. Isoproterenol (ISO) alone treated heart showed degeneration and disorganization in myocardial fibers with separation of myofibrils, Pyknotic nuclei, Cytoplasmic vacuolization of cardiac muscle fibers, edema, inflammatory cell infiltrations, and hemorrhage. These changes were remarkably reduced by pretreatment of rats with different doses of 7-HF (5, 10, and 25 mg/kg) + ISO. 7-HF reduced these changes in a dose-dependent manner. Pretreatment with the standard drug atenolol also considerably prevented myocardial damage (Figure 5).

#### 3.2.4. Effect of 7-HF on Oxidative Stress and Antioxidant Enzymes

Treatment of rats with ISO significantly elevated the end product of lipid peroxidation MDA, which is a marker of oxidative stress in heart tissues, along with the decrease in the levels of endogenous antioxidant enzymes as compared to the control group. Pretreatment of rats with 7-HF and atenolol significantly reduced the level of MDA and increased the levels of SOD, CAT, and GSH (Figure 6A–D).

#### 3.2.5. PCR Analysis of Nrf2 and iNOS

7-HF significantly up-regulated the mRNA expression of Nrf2 at 5, 10, and 25 mg/kg compared to the ISO-alone treated group, with 25 mg/kg showing more significant expression in comparison with the standard drug (atenolol). 7-HF down-regulated the mRNA expression of iNOS at 5, 10, and 25 mg/kg in contrast to the ISO-alone treated group (Figure 7).

#### 3.2.6. Effect of 7-HF on Isolated Rat Atria

The right atria of rats were used to explore the effect of 7-HF on heart rate and force of contraction. 7-HF significantly reduced heart rate at doses of (0.3, 1, 3, 10, and 30 µM) and significantly reduced the force of contraction at doses of (1, 3, 10, and 30 µM) showing its negative (–ve) chronotropic and inotropic activity (Figure 8A). Tissue pretreated with atropine (10 µM) did not cause any significant change in the effect of 7-HF on rat right atria (Figure 8B). Tissue pretreated with atenolol (0.5 µM) completely blocked the −ve chronotropic and −ve inotropic effect of 7-HF on the right atrium (Figure 8A).

#### 3.2.7. Effect of 7-Hydroxy Frullanolide on Pre-Contraction Produced by Phenylephrine, K^+^ (80 mM) and Ang II

7-HF produced a vasorelaxant effect on endothelium intact aortic ring pre-contracted with the standard vasoconstrictors i.e., phenylephrine (1 µM), high K^+^ (80 mM) and angiotensin II (5 µM) with EC_50_ value of 12.5 (09.31–14.67), 8.82 (6.52–10.73), and 13.8 μg/mL (9.93–15.26) respectively (Figure 9A), compared to the standard drug verapamil (Figure 9B).

#### 3.2.8. Endothelium-Dependent and Independent Effect of 7-Hydroxy Frullanolide

The cumulative addition of 7-HF on intact endothelium of rat aortic rings, pre-contracted with phenylephrine (1 μM), induced a relaxant effect with an EC_50_ value of 12.5 μg/mL (09.31–14.67) (Figure 9C), which was not significantly changed with the endothelium-denuded aorta EC_50_ = 13.2 μg/mL (10.51–16.18) (Figure 9C).

#### 3.2.9. Evaluating the Effect of 7-Hydroxy Frullanolide in Aorta Pre-Incubated with L-NAME, Atropine and Indomethacin

To explore the potential role of nitric oxide (NO), endothelium intact aortic tissues were pre-incubated for 15–20 min with L-NAME (10 µM). The vasorelaxant effect of 7-HF was not significantly affected {EC_50_ = 11.9 μg/mL (8.32–13.78)} with L-NAME pre-incubation as compared with the control {EC_50_ = 12.5 μg/mL (09.31–14.67)}. The tissues were also pre-incubated with atropine (1 µM) to identify the potential contribution of muscarinic receptors in the vasorelaxant effect of 7-HF. However, no significant decrease {EC_50_ = 11.6 μg/mL (8.01–13.52)} was observed in the vasorelaxant effect in the presence of muscarinic blocker, excluding the involvement of muscarinic receptors (Figure 9D).

Further, to elaborate on the association of vasorelaxant prostaglandins in the effect of 7-HF, endothelium aortic rings were pretreated with indomethacin (1 µM). The pre-incubation of tissue with indomethacin also did not show any significant change in the vasorelaxation effect of the compound {EC_50_ = 13.5 μg/mL (11.26–16.50)} (Figure 9D).

#### 3.2.10. Effect of 7-Hydroxy Frullanolide on K^+^-Channels

To study the potential involvement of K^+^ channels activation in the vasorelaxant effect of the 7-HF, K^+^ channels were pre-incubated with different inhibitors like TEA (1 mM) and BaCl_2_ (30 µM). The vasorelaxant effect of 7-HF was not significantly affected by these inhibitors, showing an EC_50_ value of 11.9 μg/mL (8.10–13.52) in the presence of TEA and an EC_50_ value of 14.1 μg/mL (11.56–16.78) in the presence of BaCl_2_ as compared with the control EC50 = 12.5 μg/mL (09.31–14.67) (Figure 9E).

#### 3.2.11. Effect of 7-Hydroxy Frulanolide on Calcium Channels

Aortic rings pre-incubated with different doses of 7-HF (3 and 10 µg/mL) significantly (*p* < 0.001) inhibited the response curve of CaCl_2_ (0.01–1 mM) in Ca^2+^-free/EGTA Kreb’s solution, shifting the CaCl_2_-CRCs towards the right, inhibiting the maximum response, like verapamil (Figure 10A,B). Compared with the control, the vehicle used for the dissolution of 7-HF (3% DMSO) did not cause any significant change in the CaCl_2_-CRCs.

#### 3.2.12. Effect of 7-Hydroxy Frulanolide on Intracellular Ca^2+^ Stores

Studies were conducted to evaluate the effect of 7-HF on phenylephrine (1 µM) induced transient contractile response. The aortic tissue was pretreated with different concentrations (1, 3, 10, and 30 µg/mL) of 7-HF. Concentration of 1 µg/mL of 7-HF did not show any significant effect, 3 µg/mL had less significant (*p* < 0.05), while 10 and 30 µg/mL significantly (*p* < 0.001) attenuated the contractile response of phenylephrine in Ca^2+^-free/EGTA Kreb’s solution, corresponding to the response produced by verapamil (0.3 and 1 µg/mL) (Figure 11A,B).

## 4. Discussion

Isoproterenol (ISO), a synthetic β-adrenergic receptor agonist, has been studied to induce cardiac injury at higher doses via different mechanisms. The chronic activation of β_1_-adrenergic receptor results in dysregulation of underlying pathways, which leads to elevation in intracellular calcium load and oxidative stress [38,39]. Persistent increase in the force and velocity of contraction of left ventricles by ISO increases oxygen demand, causing a mismatch between the supply and demand of oxygen to cardiomyocytes [40]. An increase in cardiac work also causes hypoperfusion in the coronary bed [41]. Hence, it further speeds up the process of necrosis. Changes induced by ISO correspond in different aspects to the alterations occurring in human beings after MI. Therefore, induction of MI by administration of ISO in rats has been readily used to evaluate the possible protective effect of drugs on myocardial injury [42].

Injecting high doses of ISO subcutaneously caused acute stress in cardiomyocytes of SD rats, resulting in necrosis presenting as marked elevation of ST segment and development of pathological Q-wave in the ECG comparable to that of patients presenting with Acute MI [43]. Rats treated with different doses of 7-HF and atenolol significantly reduced the elevation of ST-segment and production of pathological Q-wave in comparison with the ISO alone treated group, suggesting the protective effect of 7-HF on cardiomyocytes.

The cardioprotective effect of 7-HF was further explored by measuring the infarct area through TTC staining. TTC is reduced by mitochondrial enzyme succinate dehydrogenase of functionally active cells into photosensitive formazan that turns healthy normal tissue dark red. The infarcted tissue cannot reduce TTC, resulting in an unstained pale whitish area [44]. The same changes in a white and pale color were examined in tissues of ISO-alone treated rats showing necrosis due to non-reduction of the stain. Pretreatment of groups with 7-HF and atenolol significantly reduced the infarct size, showing its protective effect on cardiomyocytes.

MI is also clinically diagnosed by different cardiac biomarkers, upon necrosis, these markers are released into the extracellular matrix [45]. cTnI and CK-MB are highly specific and more sensitive to cardiac injury [46]. Pretreatment with different doses of 7-HF (5, 10, and 25 mg/kg) and standard drug atenolol (10 mg/kg) significantly (*p* < 0.001) decreased the level of cTnI and CK-MB, compared with the ISO alone treated group.

Other non-specific biomarkers, such as LDH, ALT, and AST, were also studied. Pretreatment of rats with 7-HF significantly reduced the level of LDH, AST, and ALT at doses of 10 and 25 mg/kg, but the low dose of 7-HF (5 mg/kg) did not cause any significant change in the serum levels as compared with ISO alone treated group. Furthermore, the histopathological evaluation of the ISO-alone treated group revealed disorganization and degeneration in myocardial fibers with separation of myofibrils, marked infarcted area, along with inflammation, edema, and hemorrhage. Pretreatment of rats with 7-HF and atenolol prior to ISO administration remarkably prohibited the pathological damage caused by ISO. 7-HF reduced the release of cardiac markers and histopathological damage potentially by the activation of Nrf2 (anti-oxidant defense system) and inhibiting expression of cytotoxic iNOS, preventing myocardial tissue against free radical-induced injury caused by ISO.

An increase in the production of free radicals after ISO administration has been associated with myocardial infarction via complex cellular and molecular pathways, and it is additionally intensified by the impairment in the endogenous antioxidant defense system [47]. Pretreatment of rats with 7-HF significantly reduced the oxidative stress by reducing the level of MDA, and significant elevation was seen in the levels of endogenous antioxidant enzymes as compared to negative control, indicating the potential protective effect of 7-HF against the ROS-induced injury caused by ISO.

To further explore the underlying mechanism responsible for the cardioprotective effect of 7-HF, its effect was investigated on the mRNA expression of Nrf2 and iNOS. Nuclear factor (erythroid-derived 2)-like 2 (NFE2L2) known as Nrf2, a cytoprotective gene that regulates the expression of different antioxidants and detoxification genes such as NQ01, HO-1 along with the increase in the level of SOD-1, catalase, and glutathione peroxidase [48]. Pretreatment of rats with 7-HF (5, 10, and 25 mg/kg) significantly (*p* < 0.001) increased the expression level of Nrf2 as compared to the negative control group, suggesting the cardioprotective effect of 7-HF by potentiation of Nrf2. Nitric oxide (NO) is a small biological molecule that is produced as a free radical by three different nitric oxide synthase isoforms in the heart: inducible (iNOS), endothelial (eNOS), and neuronal (nNOS). NO contributes to different physiological and pathological processes. The physiological NO is generated by eNOS for a short duration (secs or mints), while the cytotoxic NO is produced by iNOS for a large duration (hours or days) [49]. Pretreatment of rats with 7-HF significantly (*p* < 0.001) reduced the expression of iNOS, hence preventing the extent of damage caused by ISO.

Further experiments were performed to evaluate the other possible mechanisms responsible for the cardioprotective activity of 7-HF. Increased heart rate and peripheral vascular resistance are important factors of myocardial infarction. The effect of 7-HF on the heart rate and force of contraction was explored using the right atrium of SD rats. 7-HF significantly reduced the rate and force of contraction of the heart, which was completely inhibited in the presence of atenolol (β1 adrenergic receptor blocker), suggesting its potential effect on the β1 adrenergic receptor. To evaluate the effect of 7-HF on peripheral vascular resistance, an isolated rat aorta was used. 7-HF produced significant relaxation of aortic rings pre-contracted with standard vasoconstrictors such as phenylephrine (1 µM), high K^+^ (80 mM), and angiotensin II (5 µM). To confirm the possible involvement of endothelium in the vasorelaxant effect of 7-HF, its effect was evaluated on endothelium-dependent vasorelaxation pathways, including synthesis of NO, activation of PLC and IP_3_ through muscarinic receptor and release of prostacyclin [29].

Aortic tissue pre-incubated with the inhibitors of these pathways, L-NAME, atropine, and indomethacin failed to inhibit the vasorelaxant property of 7-HF, which suggests that it may act through other pathways involving smooth muscles [50]. To assist this possibility, the effects of 7-HF against different vasoactive substances, such as high K^+^ and angiotensin II, were examined.

In comparison with the standard drug verapamil, 7-HF produced a significant vasorelaxant effect in aortic rings pre-stimulated with high K^+^. Contraction induced by high K^+^ is mediated by the influx of Ca^2+^ into the cells through voltage-dependent calcium channels (VDCCs) [51,52]. It is evident that a substance that inhibits precontraction caused by high K^+^ levels could potentially prevent Ca^2+^ entry into the cell [7]. These findings imply that the vasorelaxant effect of 7-HF may be mediated by the inhibition of Ca^2+^ entry through VDCCs. Further experiments were performed to explore this possibility. CaCl_2_-CRCs were obtained by suspending the aortic rings in an organ bath containing Ca^2+^-free/EGTA medium. 7-HF suppressed the maximum response of CaCl_2_ in comparison to the standard drug, verapamil. These data reveal that 7-HF significantly inhibits the Ca^2+^ entry through VDCCs.

As examined previously, 7-HF caused a vasorelaxant effect against phenylephrine-induced contraction, suggesting the potential inhibitory action of 7-HF on the release of calcium from intracellular Ca^2+^ stores (sarcoplasmic reticulum). 7-HF significantly inhibited phenylephrine-induced contraction in a concentration-dependent manner compared to the control, indicating that 7-HF inhibits the release of Ca^2+^ from intracellular stores in the same mechanism as that of verapamil, a Ca^2+^ channel blocker [31]. These findings prompted us to conduct further experiments to examine if 7-HF influenced Ca^2+^ moments via membrane-associated channels.

In comparison to verapamil, 7-HF produced vasorelaxation against high k-induced precontact at higher concentrations, implying that it may affect other vascular channels. Another possible mechanism is the stimulation of potassium channels because vasodilators that rely on K^+^ channel activation lose their activity when exposed to high K^+^ [53]. To explore the potential involvement of K^+^ channels in the vasorelaxant response of 7-HF, aortic rings were pre-incubated with TEA (non-selective blocker) and BaCl_2_ (selective blocker). No significant change in the vasorelaxant effect of 7-HF was examined, excluding the involvement of K^+^ channels.

## 5. Conclusions

In conclusion, the current study has demonstrated an important role of 7-HF in protecting heart against ischemic injury by suppressing oxidative stress and inflammation due to the modulation of Nrf2 and iNOS genes and improving cardiac function by the inhibition of voltage-dependent calcium channels (decreasing calcium overload), indicating 7-HF a promising effective agent for MI. However, further studies are required to confirm these effects by western blotting and patch clamping. In addition, the long-term beneficial effect of 7-HF on cardiac function, such as ejection fraction (EF) and left ventricular volume and pressure has to be evaluated.

## Figures and Tables

**Figure 1 biomedicines-11-02470-f001:**
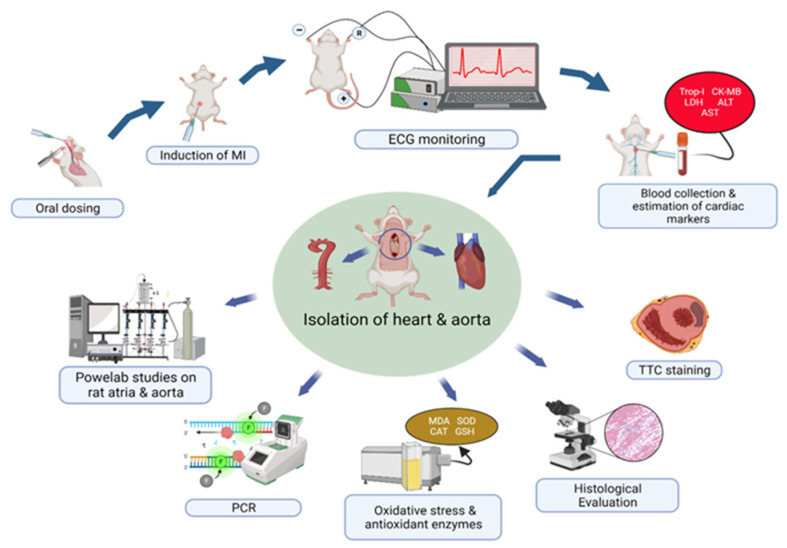
Research protocol of cardioprotective activity of 7-hydroxy frullanolide against isoproterenol induced myocardial infarction.

**Figure 2 biomedicines-11-02470-f002:**
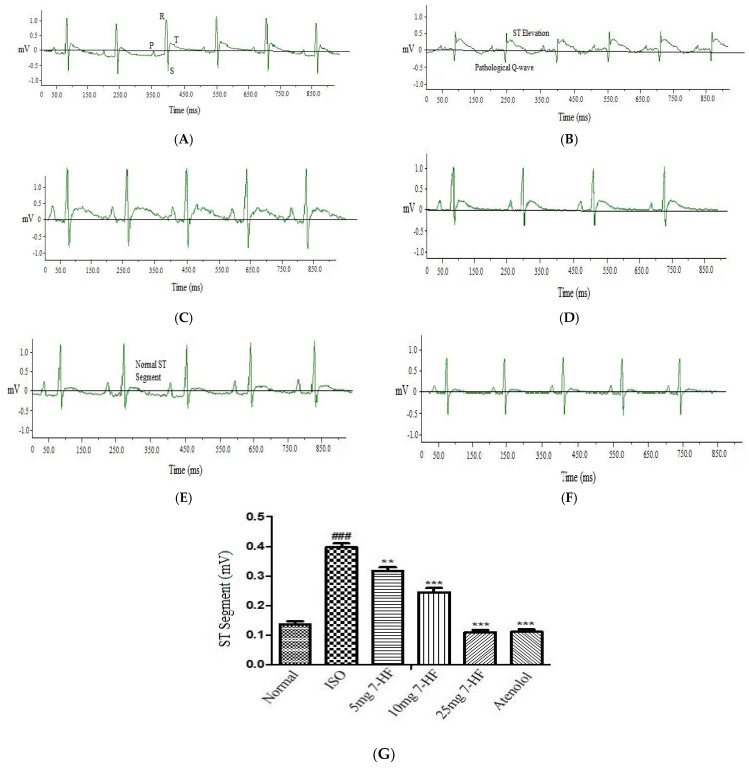
Represents electrocardiogram (ECG) lead II tracing of (**A**) control showing normal P wave, RS pattern and T wave (**B**) isoproterenol (ISO)-alone-treated group showed pathological Q wave and marked ST-segment elevation. Representative electrocardiogram (ECG) lead II tracing of 7-HF treated groups (**C**) 5 mg/kg showed complete disappearance of pathological Q wave and decrease in ST-segment elevation (**D**) 10 mg/kg further reduced ST elevation (**E**) 25 mg/kg showed almost normal ECG pattern, (**F**) shows effect of atenolol on ECG pattern (Recorded at speed of 50 ms/div). ms = millisecond, mv = millivolt, and (**G**) shows the graphical representation of the effect of 7-HF on ST-segment elevation. Values are expressed as Mean ± SEM (n = 5), analyzed by one way analysis of variance (ANOVA) followed by post hoc comparison with Tukey’s test. Where ** *p* < 0.05 vs. ISO, *** *p* < 0.001 vs. ISO, ### *p* < 0.001 vs. Normal.

**Figure 3 biomedicines-11-02470-f003:**
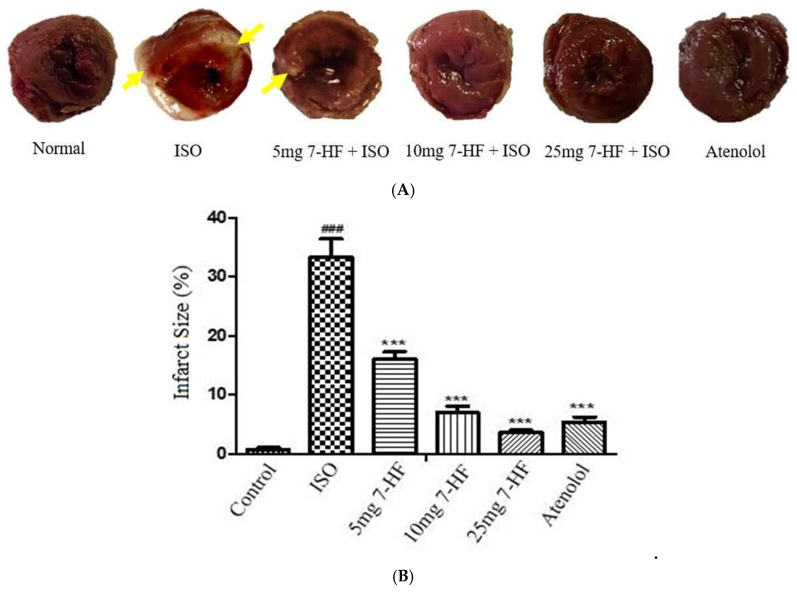
Shows heart tissue stained with TTC. Brick red (dark region) in the normal group indicated mitochondrial aspiration. ISO-alone treated group shows a large white region (arrow) indicating the infarcted tissue (**A**). Graph shows the effect of 7-HF on infarct size in ISO-induced MI. The infarct size is expressed as % area to the total ventricular area (**B**). Values are expressed as Mean ± SEM (n = 4), analyzed by one-way analysis of variance (ANOVA) followed by post hoc comparison by Tukey’s test. Where *** *p* < 0.001 vs. ISO, ^###^
*p* < 0.001 vs. control.

**Figure 4 biomedicines-11-02470-f004:**
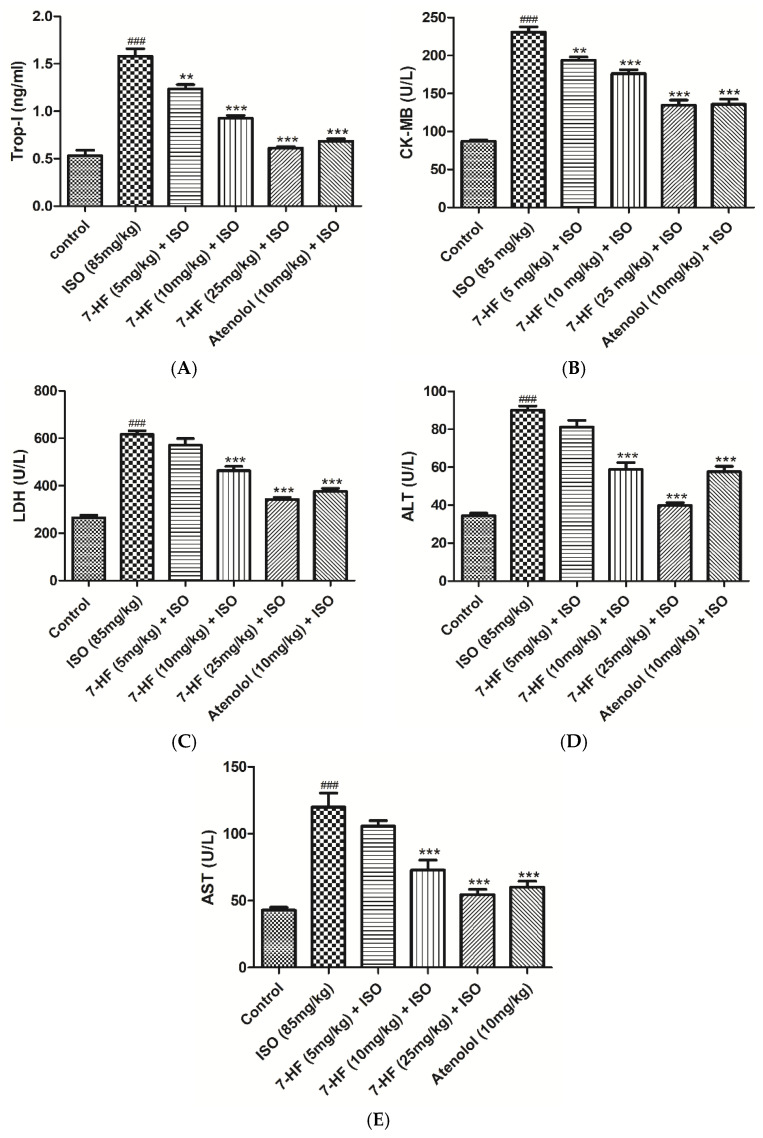
Represents the effect of 7-HF and atenolol on cardiac troponin I (cTnI) (**A**), creatine kinase-MB (CK-MB) (**B**), lactate dehydrogenase (LDH) (**C**), alanine transaminase (ALT) (**D**), and aspartate transaminase (AST) (**E**) in isoproterenol (ISO)-induced ischemic rats. Values are expressed as Mean ± SEM, analyzed by one-way analysis of variance (ANOVA) followed by post hoc comparison by Tukey’s test. Where ** *p* < 0.01 vs. ISO, *** *p* < 0.001 vs. ISO, ^###^
*p* < 0.001 vs. control.

**Figure 5 biomedicines-11-02470-f005:**
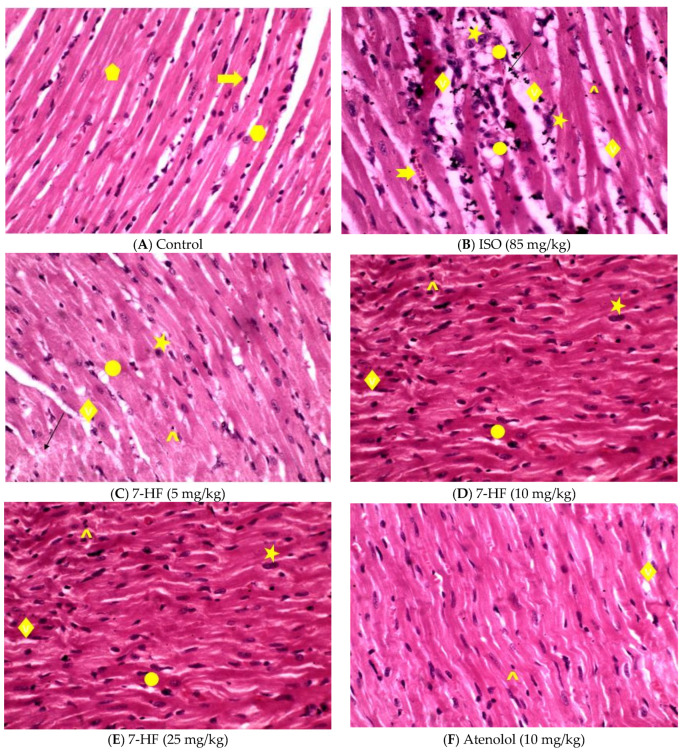
H&E staining of heart tissue (40×) (**A**) Control, shows normal cardiomyocytes with vesicular central oval nuclei (hexagon), acidophilic cytoplasm (pentagon), and the cardiac fibers isolated by interstitial connective tissue with flat nuclei (arrow). (**B**) Isoproterenol (ISO)-alone treated heart shows degeneration and disorganization in myocardial fibers with separation of myofibrils (thin arrow), pyknotic nuclei (star), cytoplasmic vacuolization of cardiac muscle fibers (circle), edema (diamond), inflammatory cells infiltrations (caret) and hemorrhage (notched arrow). These changes were remarkably reduced by pretreatment with different doses of 7-HF + ISO (5, 10 and 25 mg/kg) (**C**–**E**) and the standard drug atenolol (**F**). 7-HF reduced these changes in a dose-dependent manner with the highest dose 25 mg/kg showing a more significant reduction in ISO-induced myocardial damage. Pretreatment with the standard drug atenolol also considerably reduced myocardial damage.

**Figure 6 biomedicines-11-02470-f006:**
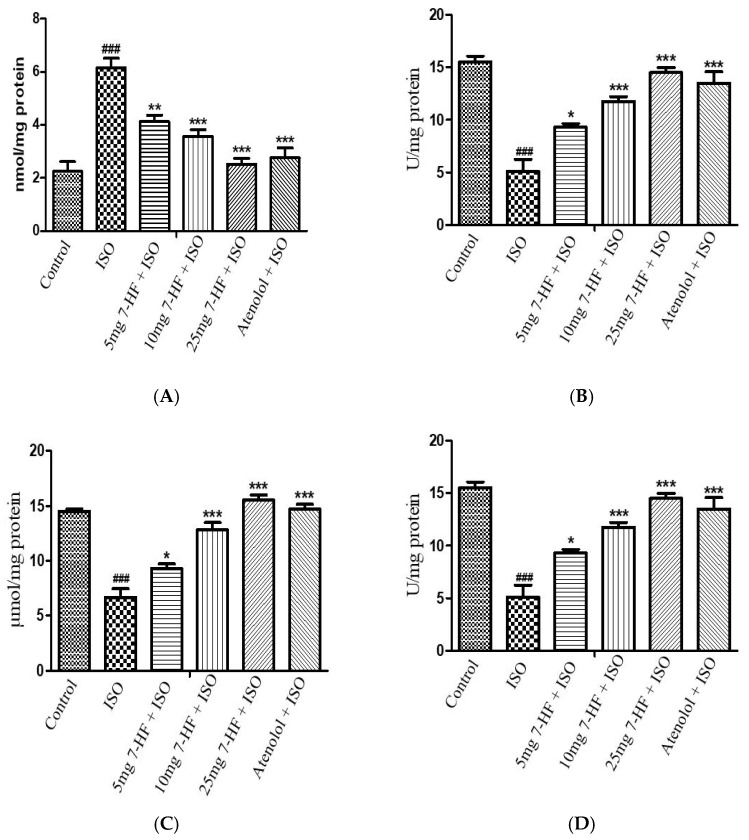
Shows the effect of 7-HF and atenolol on the level of malondialdehyde (**A**), superoxide dismutase (**B**), catalase (**C**), and reduced glutathione (**D**) in heart homogenates. Values are expressed as Mean ± SEM, analyzed by one-way analysis of variance (ANOVA) followed by post hoc comparison by Tukey’s test. Where * *p* < 0.05 vs. ISO, ** *p* < 0.01 vs. ISO, *** *p* < 0.001 vs. ISO, ^###^
*p* < 0.001 vs. control.

**Figure 7 biomedicines-11-02470-f007:**
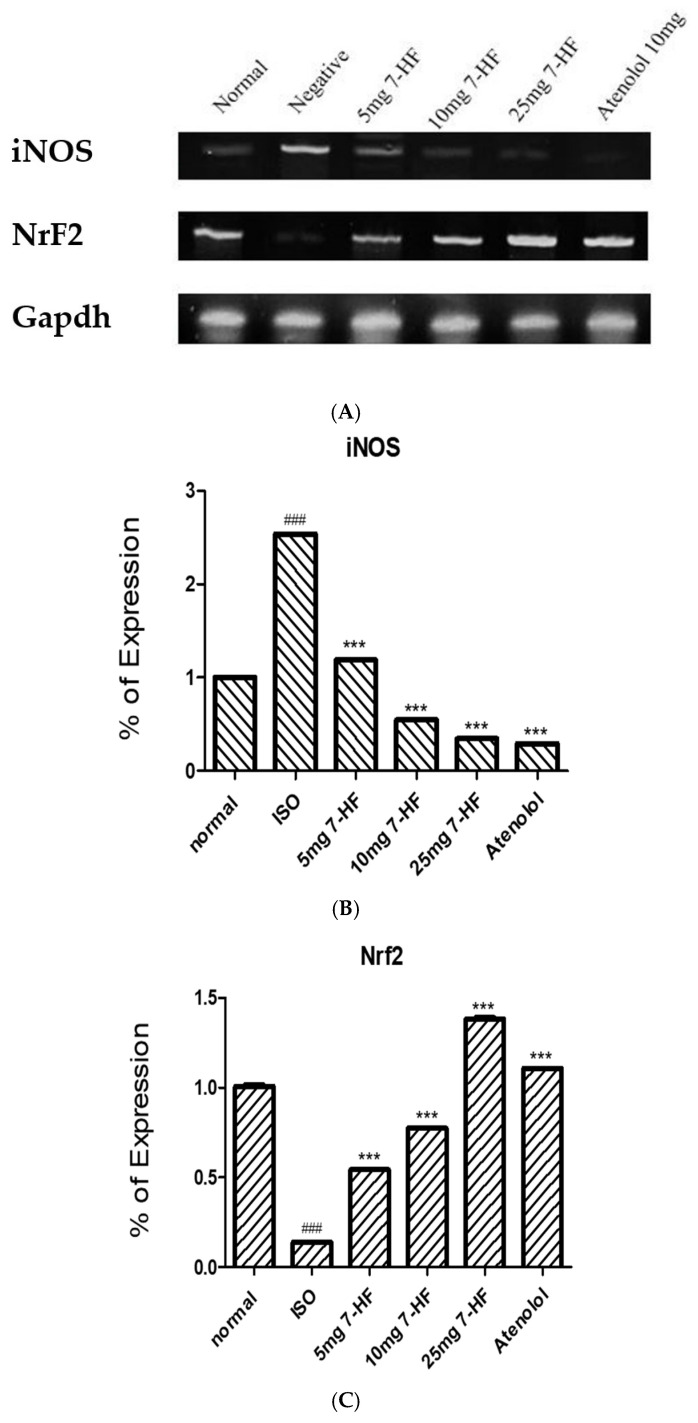
Shows the images (**A**) and graphs of PCR analysis of Nrf2 and iNOS in heart tissues. 7-HF significantly decreased the mRNA expression of iNOS (**B**) and significantly increased the mRNA expression of Nrf2 (**C**) as compared to the ISO-alone treated group. Values are shown as mean ± SEM. **^###^**
*p* < 0.001 vs. normal, *** *p* < 0.001 vs. ISO.

**Figure 8 biomedicines-11-02470-f008:**
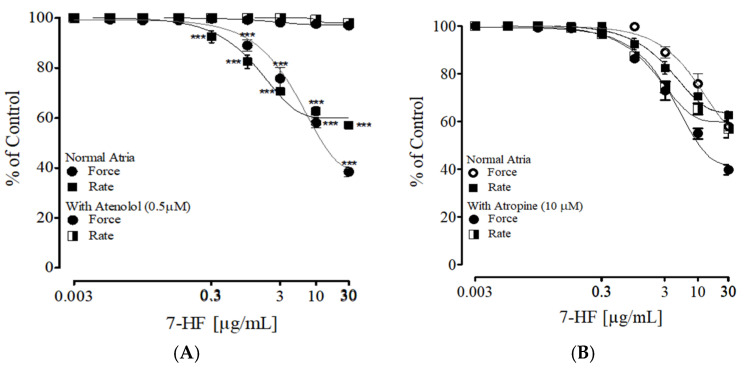
Graph shows the effect of 7-HF on the force and rate of contraction on spontaneous atrium as well as tissue pre-incubated with atenolol (0.5 µM) (**A**) and atropine (10 µM) (**B**). Two-way ANOVA followed by Bonferroni post hoc analysis was applied to determine any statistical difference between the variables. (n = 5). Where *** *p* < 0.001 vs. Atenolol.

**Figure 9 biomedicines-11-02470-f009:**
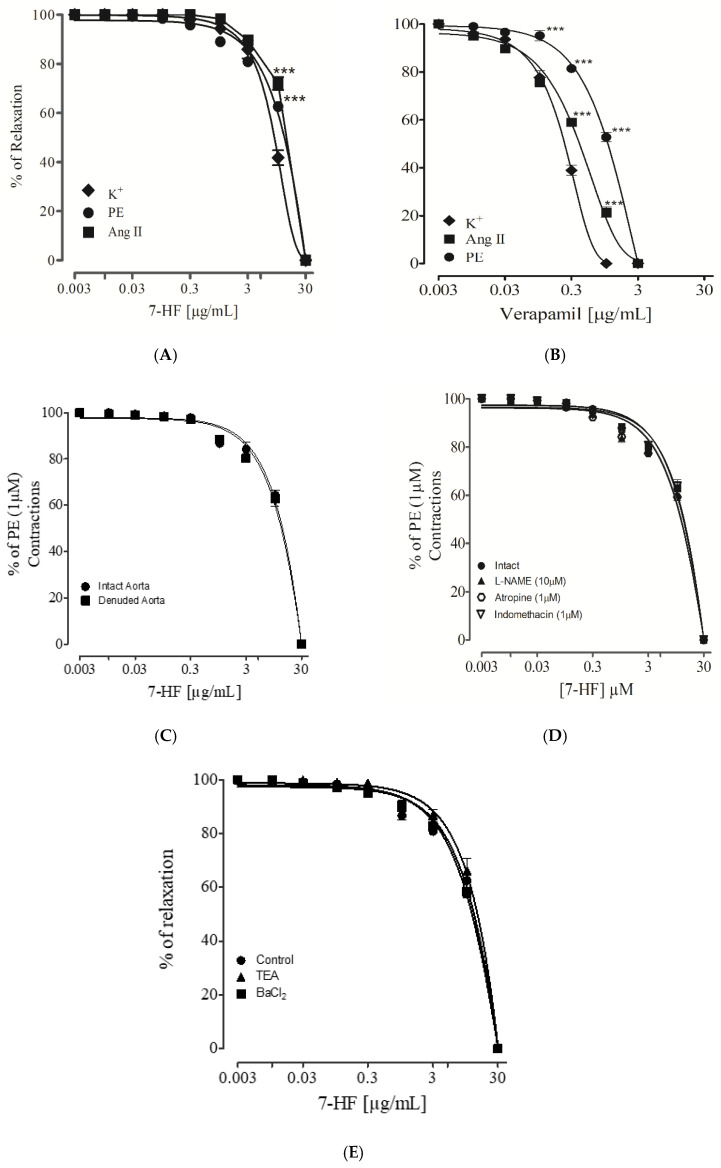
Graphs showing the effect of 7-HF and verapamil against standard vasoconstrictor i.e., phenylephrine (PE), K^+^ (80 mM) and Ang II (**A**,**B**), endothelium-intact and denuded tissue (**C**), pretreated; L-NAME (10 µM) atropine (1 µM) and indomethacin (1 µM) (**D**), pretreated with tetraethylammonium (TEA; 1 mM) and Bacl_2_ (30 µM) (**E**). The relaxation responses are expressed as mean± SEM for five experiments. Two-way ANOVA followed by Bonferroni post hoc analysis was applied to determine any statistical difference between the variables. Where *** *p* < 0.001 vs. Control.

**Figure 10 biomedicines-11-02470-f010:**
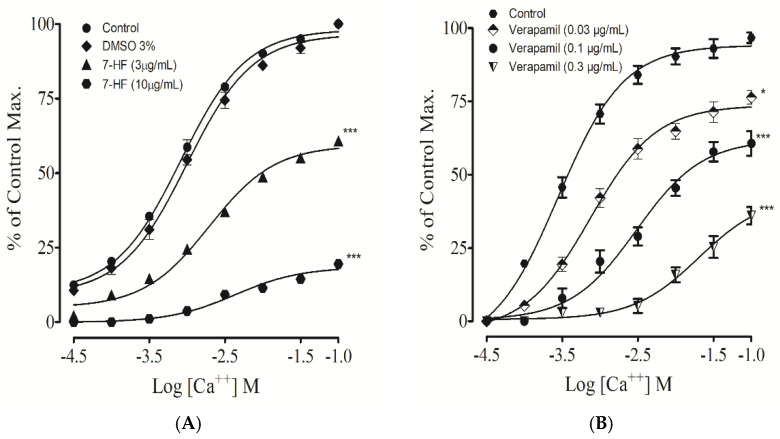
The graph illustrates the effect of 7-HF (**A**) and verapamil (**B**) on Ca^2+^ response curve produced in Ca^2+^—free/EGTA solution, in isolated rat aorta. The relaxation responses were presented as mean ± SEM for five experiments. Where * *p* < 0.05 vs. Control, *** *p* < 0.001 vs. Control, represents the significant difference.

**Figure 11 biomedicines-11-02470-f011:**
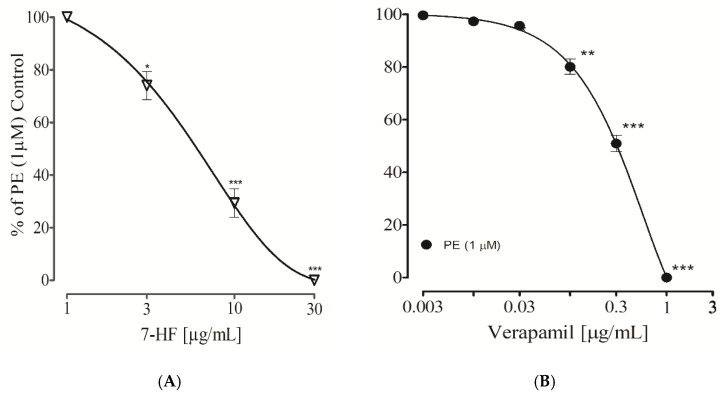
The graphs show the effects of increasing concentrations of 7-HF (**A**) and verapamil (**B**) on peaks of contractions produced by phenylephrine (PE) in Ca^2+^-free/EGTA solution. The relaxation responses were presented as means ± SEM for five experiments. Where * *p* < 0.05 vs. Control, ** *p* < 0.01 vs. Control, *** *p* < 0.001 vs. Control.

**Table 1 biomedicines-11-02470-t001:** Primers sequence design for PCR assessment.

Accession No	Genes	Primer Sequence (5′-3′)
NM_031789.2	Nrf2	F-TTGTAGATGACCATGAGTCGC R-ACTTCCAGGGGCACTGTCTA
XM_039085203.1	iNOS	F-GCAGGGCCACCTCTATGTTT R-TGGTCACCCAAAGTGCTTCA
NM_017008.4	Gapdh	F-GGGTGTGAACCACGAGAAAT R-ACTGTGGTCAATGAGCCCTTC

Nrf2 = Nuclear factor erythroid 2–related factor 2, iNOS = Inducible nitric oxide synthase. Gapdh = Glyceraldehyde 3-phosphate dehydrogenase.

## Data Availability

The data presented in this study are all available in the manuscript.

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
