# Peer review of "7-Hydroxy Frullanolide Ameliorates Isoproterenol-Induced Myocardial Injury through Modification of iNOS and Nrf2 Genes"

_biomedicines, 2023, doi:10.3390/biomedicines11092470_

Round 1

Reviewer 1 Report

Review of Manuscript: biomedicines-2547309

Title: 7-Hydroxy Frullanolide Ameliorates Isoproterenol-induced Myocardial Injury through Modification of iNOS and Nrf2 genes

Wu et al. designed a study to explore the potential anti-inflammatory effects of 7-hydroxyfrullanolide (7-HF) in the treatment of myocardial infarction. They utilized a mouse model of isoproterenol (ISO)-induced MI to determine the cardioprotective effects of 7-HF on infarct size, structure and release of cardiac biomarkers. 7-HF reduced cardiac workload upon reducing peripheral vascular resistance and enhanced cardiac Nuclear factor (erythroid-derived 2)-related factor 2 (Nrf2) and reduced iNOS expression. Moreover, 7-HF inhibited of voltage-dependent calcium channels, thereby releasing intracellular calcium stores in vitro. The manuscript is well-written and easy to follow.

NOVELTY:

The anti-inflammatory and apoptotic effects of 7-HF have been demonstrated in experimental models of acute and chronic inflammation (PMID: 20621086, PMID: 21296061) and cancer (PMID: 30535334), respectively. Moreover, methanolic extract of S. indicus, of which the active ingredient is 7-HF, exerted anti-atherosclerotic effects in a high fat-fed LDLr-/- mouse and a high fat-fed hyperlipidemic hamster (PMID: 26064179). Moreover, the authors offer ex vivo models, such as isolated aortic rings, in order to demonstrate the effects of 7-HF against vasoconstriction and Ca2+ channels and isolated right atria to explore the potential effect on β1 adrenergic receptor.

Major comments

1.     The design of the study is preventative, given that 7-HF was administered for 8 days prior to ISO-induced MI. What is the translational potential of the cardioprotective effects of 7-HF in this context?

2.     Was TTC staining quantification performed at various layers of cardiac tissue?

3.     Results, line 332. The GSSG/GSH ratio is often used to quantify degree of oxidative stress. In the current study, the authors measured GSH levels. Were GSSG levels also determined?

4.     The authors elegantly show that beneficial effects of 7-HF occur in association with reduced levels of oxidative stress. However, whether 7-HF exerts cardioprotective effects through ROS scavenging or detoxification could be demonstrated through culturing isolated rat aorta, atrium and/or cardiomyocytes with ROS-producers and/or inhibitors of ROS detoxifying enzymes.

5.     The authors mention the downstream NRF2 targets, NQ01, HO-1. Were these measured in the current study? Moreover, ex vivo experiments would benefit from use of the NRF2-specific pharmacological inhibitor ML385, for example, in order to better support this possible mechanism.  

6.     The authors demonstrate a vasorelaxant effect of 7-HF on isolated aortic rings, upon inhibitsing the Ca2+ entry through voltage-dependent calcium channels (VDCCs). In cardiac and smooth muscle cells the predominant VDCCs are the dihydropyridine-sensitive CaV1.2 channels (also called L-type channel). In order to link the ex vivo findings to the in vivo results of improved ISO-induced alterations in ECG pattern, it would be beneficial to measure the expression of CaV1.2 channels in the hearts.

Minor comments

1.     Introduction, line 66: Please define 7-HF at first mention, then use abbreviation throughout (line 75-76)

2.     Methods, line 169: “The sequence of primers for specific genes is listed in Table 2.1.” Please correct to Table 1.

3.     Results, line 28: Please correct to “As shown in Figure 4.” Moreover, reporting of values for cardiac biomarkers in text is redundant for those represented graphically in Figure 4.

4.     Please include n numbers for all experiments in Figure legends (particularly ex vivo experiments).  

Author Response

Response to Reviewer Comments

Title: 7-Hydroxy Frullanolide Ameliorates Isoproterenol-induced Myocardial Injury through Modification of iNOS and Nrf2 genes  

We are thankful to the reviewers for their fruitful suggestions and comments. A point to point response to their comments is provided below while the manuscript is thoroughly revised and improved. The changes made in the revised manuscript are highlighted in red color text.

Reviewer #1:

Comment. The design of the study is preventative, given that 7-HF was administered for 8 days prior to ISO-induced MI. What is the translational potential of the cardioprotective effects of 7-HF in this context?

Response: In our study, we have used the preventive protocol to reduce the myocardial injury and improved cardiac function, However the effect of 7-HF on long term mortality benefits and on prognosis is out of the domain of the current study.

Comment. Was TTC staining quantification performed at various layers of cardiac tissue?

Response: For the performance of TTC staining the left ventricle was sectioned into transverse slices in a plane parallel to the atrioventricular groove, containing all layers of cardiac tissue. TTC staining is performed for both transmural infarction (involving all layers of cardiac tissue) as well as non-transmural infarction. For the calculation of layers infarcted an index called transmural extension index, TME is used which measures the degree of extension of the infarct from the endocardium toward the epicardium. However, in over study, we have calculated the overall infarct size as % area infarcted to the total ventricular area, without calculating the TME.

Comment.  Results, line 332. The GSSG/GSH ratio is often used to quantify degree of oxidative stress. In the current study, the authors measured GSH levels. Were GSSG levels also determined?

Response: The reduced (GSH)-to-oxidized (GSSG) glutathione ratio represents a dynamic balance between oxidants and antioxidants. However, in our study we have determined the lipid peroxidation as a marker of oxidative stress. Animal and human studies both support a potential role of lipid peroxidation in predicting the progression of CVD and response to therapies. The most studied maker of lipid peroxidation is malondialdehyde (MDA), the levels of which was determined in heart homogenates in our study. And for the antioxidant activity the effect of 7-HF was explored on the levels of antioxidant enzymes such SOD, CAT and GSH. We have only determined the level of reduced glutathione (GSH) due to its inhibitory action on lipid peroxidation.

Comment. The authors elegantly show that beneficial effects of 7-HF occur in association with reduced levels of oxidative stress. However, whether 7-HF exerts cardioprotective effects through ROS scavenging or detoxification could be demonstrated through culturing isolated rat aorta, atrium and/or cardiomyocytes with ROS-producers and/or inhibitors of ROS detoxifying enzymes.

Response: Thank you for your valuable suggestion. We acknowledge that demonstration of ROS scavenging property of 7-HF through culturing isolated cardiomyocytes will further evaluate the the anti-oxidant property of 7-HF. However, currently we do not have such facilities in our lab. We will definitely consider it in our future work.

Comment. The authors mention the downstream NRF2 targets, NQ01, HO-1. Were these measured in the current study? Moreover, ex vivo experiments would benefit from using the NRF2-specific pharmacological inhibitor ML385, for example, to better support this possible mechanism.  

Response: We have only explored the effect of 7-HF on NRF2 without exploring its downstream targets which is a limitation of our study. We highly acknowledge reviewer suggestions about ex vivo experiments on the NRF-2 pathway using its pharmacological inhibitor ML385. However, currently we do not have such a facility in our lab we will must consider it in our future studies.

Comment. The authors demonstrate a vasorelaxant effect of 7-HF on isolated aortic rings, upon inhibiting the Ca2+ entry through voltage-dependent calcium channels (VDCCs). In cardiac and smooth muscle cells the predominant VDCCs are the dihydropyridine-sensitive CaV1.2 channels (also called L-type channel). In order to link the ex vivo findings to the in vivo results of improved ISO-induced alterations in ECG pattern, it would be beneficial to measure the expression of CaV1.2 channels in the hearts.

Response: Thank you for your valuable suggestion, we will consider it in our future studies.

Minor Comments

  1. Introduction, line 66: Please define 7-HF at first mention, then use abbreviation throughout (line 75-76)

Response:  Thank you, we followed this suggestion.

  1. Methods, line 169:“The sequence of primers for specific genes is listed in Table 2.1.” Please correct to Table 1.

Response:  Correction is made, thank you

  1. Results, line 28:Please correct to “As shown in Figure 4.” Moreover, reporting of values for cardiac biomarkers in text is redundant for those represented graphically in Figure 4.

Response: Thank you. We have added this sentence.

  1. Please include n numbers for all experiments in Figure legends (particularly ex vivoexperiments).  

Response: The number of experiments is mentioned in the legends for ex vivo experiments. For other studies, the number of experiments was five as each group in oral dosing was consist of five rats mentioned in animal and experimental design.

Reviewer 2 Report

The manuscript entitled "7-Hydroxy Frullanolide Ameliorates Isoproterenol-induced Myocardial Injury through Modification of iNOS and Nrf2 genes" brings in attention a new compound that could potentially involved in cardioprotection in myocardial ischemia. 

The following observation has to be made:

Introduction

Please offer more details about the aim of the study (eventually write few details about the parameters you assessed and what was the aim of assessing it). 

Methods - explain the method of rat euthanasia in detail (line 177).

Please indicate the limitation of the study at the end of manuscript.

Author Response

Title: 7-Hydroxy Frullanolide Ameliorates Isoproterenol-induced Myocardial Injury through Modification of iNOS and Nrf2 genes  

We are thankful to the reviewers for their fruitful suggestions and comments. A point-to-point response to their comments is provided below while the manuscript is thoroughly revised and improved. The changes made in the revised manuscript are highlighted in red color text.

The following observation has to be made:

Comment #01: Introduction

Please offer more details about the aim of the study (eventually write few details about the parameters you assessed and what was the aim of assessing it).

Response: The introduction section was revised and improved. However, the details about each parameter assessed along with the reason is mentioned in the discussion section.

Comment #02: Methods - explain the method of rat euthanasia in detail (line 177).

Response: We mentioned it in detail in methods, thank you

Comment #03: Please indicate the limitation of the study at the end of the manuscript.

Response: We have added the limitation of the study at the end of the manuscript.